# Prevalence and incidence of decreased glomerular filtration rate and its variation over 6 years: Cohort study SABE 2010–2016

**Camila de Souza dos Santos**[1], **Yeda Aparecida de Oliveira Duarte**[2], **Dirce Maria Trevisan Zanetta**[1]*

1 School of Public Health of the University of São Paulo, São Paulo, Brazil, 2 Nursery School of the University of São Paulo, São Paulo, Brazil

* dzanetta@usp.br

## Abstract

The aging process and the rising prevalence of Chronic Noncommunicable Diseases (NCDs) contribute to the decline in kidney function among elderly individuals. The aim of this research was to assess prevalence and incidence of decreased glomerular filtration rate (GFR) (GFR <60mL/min/1.73m2) over six-year period in elderly residents of São Paulo. This study relied on data from 2010 and 2016 waves of the cohort SABE Study - Health, Wellbeing, and Aging, with a probabilistic and representative sample of elderly individuals residing in São Paulo. GFR was calculated using the 2021 Chronic Kidney Disease Epidemiology Collaboration creatinine (CKD-EPI) equation. Categorical variables were analyzed using chi-square test with Rao-Scott correction, and weighted means and standard errors were calculated for continuous variables. Logistic and linear regression models were constructed to analyse the data. Statistical analyses accounted for sample weights to ensure population representativeness. The prevalence of decreased GFR in 2010 was 17.3%, with mean GFR of 75.6 mL/min/1.73m$^2$ (SE = 0.5). The incidence of decreased GFR between 2010 and 2016 was 14.9%, equivalent to an annual incidence of 2.5%. This incidence was associated with older age, hypertension, self-perceived fair/poor/very poor health, and greater number of comorbidities associated. Over the study period, 68.1% of the elderly participants experienced deterioration in GFR, with an average decline of 1 mL/min/1.73m$^2$ each year. Renal function decline often occurs without noticeable symptoms, and the high prevalence of comorbidities contributes to the worsening of GFR. Therefore, monitoring renal function in the elderly is crucial for effectively managing the health of this population.

## Introduction

Chronic kidney disease (CKD) affects a substantial number of individuals worldwide, with an estimated prevalence of nearly 500 million people, particularly concentrated in less developed countries [1, 2]. However, the lack of population-based studies with consistent assessments of renal function hampers the accurate estimation of decreased glomerular filtration rate (GFR)

**Funding:** This retrospective analysis was conducted through the financial support provided by the São Paulo Research Foundation, FAPESP, under the grant numbers 2009/53778-3 and 2014/50649-6. Additionally, CSS was the recipient of a scholarship under CAPES Program - DS (Programa de Demanda Social, Process Number 88882.378283/2019-01).

**Competing interests:** The authors have declared that no competing interests exist.

and CKD prevalence in the population. Developing countries, including Brazil, experience higher rates of CKD due to factors such as population aging and the increased prevalence of chronic diseases [2].

The decline in GFR associated with aging is considered a natural process. Nonetheless, common chronic conditions in elderly patients, such as systemic arterial hypertension (SAH) and diabetes mellitus (DM), can contribute to further deterioration of renal function and the development of CKD [1–4].

The objective of this study is to estimate the prevalence and incidence of decreased glomerular filtration rate and its temporal variation among elderly individuals in the city of São Paulo from 2010 to 2016. To the best of our knowledge, no previous Brazilian population-based study has assessed longitudinally the GFR in the elderly population.

## Methods

### Study design

The SABE Study, initiated in 2000, utilized a representative sample of the elderly population (aged 60 years and above) residing in São Paulo city, Brazil [5]. The sample selection employed a two-stage cluster sampling method: the first stage involved the random selection of census tracts proportional to the number of households, and the second stage involved selecting households where 1568 elderly individuals were interviewed [6]. In each subsequent wave (2006, 2010, and 2016), surviving participants were re-interviewed, and a new cohort was introduced, consisting of individuals aged 60 to 64 years, as this age group was no longer represented. For this study, data from the 2010 and 2016 waves were utilized, as biochemical testing was introduced only from the 2010 wave onwards.

Data collection involved interviews, physical assessments, and blood and urine sample collection, all conducted at the participants' homes. An interviewed-administred questionnaire, consisting of 13 sections, was conducted. When cognitive impairment, evaluated by the Mini-Mental State Examination (MMSE), was present, the questionnaire was responded by an auxiliary respondent.

Sample blood and urine collection was performed at the participants' homes by nursing technicians between 7 am and 9 am, after at least ten hours of fasting. The samples were stored in thermal containers and transported to the laboratory of Instituto do Coração (INCOR/USP), which holds ISO 9001 certification. Measurement of protein was conducted using a 3% sulfosalicylic acid assay, while creatinine levels were determined using the Jaffé method, which was traceable to the "Definitive Method by Mass Spectrometry of Isotopic dilution" (IDMS).

### Variables

The glomerular filtration rate (GFR) was estimated using the 2021 Chronic Kidney Disease Epidemiology Collaboration (CKD-EPI) creatinine equation, which incorporates age and sex factors, while race was excluded. A GFR value less than 60 mL/min/1.73m$^2$ was considered decreased GFR.

Sociodemographic variables were assessed, including age, sex, education level, and living arrangements (alone, with a partner or a family member). Perception of health status was classified as either very good/good or fair/poor/verypoor, based on responses to the question "Would you say your health is . . .". The presence of chronic diseases was self-reported through the question "Has a doctor or nurse ever told you that you have/had . . .?".

The classification of hypertension was based on self-reporting or a mean of the last two of three systolic blood pressure measurements $\geq$ 140 mmHg or diastolic blood pressure $\geq$ 90 mmHg [7]. Diabetes status was determined by self-reporting or fasting blood glucose

levels $\geq$ 126 mg/dL or glycated hemoglobin levels $\geq$ 6.5% [8]. Self-reporting was utilized to assess other diseases, including heart disease, lung disease, stroke, and joint disease.

Metabolic syndrome was defined by the unified criteria proposed by several major organizations criteria in a Joint Interim Statement [9].

Quality of life (QoL) was assessed using the 12-Item Short-Form Health Survey (SF-12), classified as good (above the median score) or poor (below the median).

Physical activity (PA) was evaluated using the International Physical Activity Questionnaire - IPAQ - short version, which has been validated for use in Brazil [10]. Individuals who engaged in at least 150 minutes of moderate and/or vigorous PA per week were categorized as active, following the criteria established by the World Health Organization [11].

## Statistical analysis

The analysis incorporated the weights of individuals to adjust for varying selection probabilities among participants, ensuring the results' representativeness for the population of São Paulo city aged 60 years or older. The prevalence of decreased GFR was determined by dividing the weighted number of elderly individuals with GFR $<$60 mL/min/1.73m$^2$ in 2010 by the total weighted number of elderly individuals evaluated in that year. The incidence was calculated by dividing the weighted number of new cases with GFR $<$60 mL/min/1.73m$^2$ in 2016 among those individuals who had GFR $\geq$60 mL/min/1.73m$^2$ in 2010 divided by the weighted number of them.

Among the elderly individuals who had both GFR measurements available, the variation in GFR during the follow-up period was assessed, subtracting the GFR value in 2010 from the GFR value in 2016. The variable representing GFR variation was analyzed both as a continuous variable and categorized into two groups: those who maintained or improved GFR and those who experienced worsening, indicated by a negative difference between GFR values.

Categorical variables were analyzed using the chi-square test with Rao-Scott correction, accounting for the survey design. Continuous variables were summarized using weighted means (standard erros–SE).

A logistic regression model and a linear regression model were employed to analyze the data. The variables included in the models were selected using the Stepwise method (Backward Selection), where independent variables with a p-value $<$0.20 in the univariate analysis and those deemed important based on the literature were added to the multiple model.

In the logistic regression models, the results were reported as odds ratios (OR) with corresponding 95% confidence intervals (CI), and significance of variables was evaluated by Wald test. The goodness of fit of the final model was assessed using the Hosmer-Lemeshow test. For the linear regression model, the model selected was the one with the best fit for the residuals.

Statistical analyses were performed using the Stata/SE 13.0 for Windows program (Stata Corp., College Station, United States). The study design and the sample complexity were taken into account, employing appropriate tests and statistical analyses suitable for survey-type studies ("svy"). Sample weights were applied to ensure the representativeness of the elderly population in the city of São Paulo. The results are presented as weighted numbers.

## Ethical considerations

The SABE Study (Health, Wellbeing, and Aging) obtained approval from the Research Ethics Committee of the Faculty of Public Health, University of São Paulo (COEP) for all waves of data collection. Prior to participation, all elderly individuals or their guardians were required to provide informed consent by signing a Free and Informed Consent Form (TCLE).

## Results

### Population description and decreased GFR prevalence

A total of 1,344 elderly individuals were initially interviewed in 2010, out of which 1,259 had blood samples collected and 1,001 had urine samples collected. The sample of 1,259 participants was representative of the estimated 1,256,673 elderly residents in the city of São Paulo. The majority of the sample consisted of women (59.9%), individuals aged between 60 and 69 years (54.2%), with a mean age of 70.5 years (SE = 0.2). In terms of education, most participants had 4 to 7 years of schooling (37.1%), and lived with someone (84.7%).

Among the elderly participants, the most prevalent comorbidity was arterial hypertension (78.8%), followed by diabetes (29.5%). Approximately 54% of the participants had metabolic syndrome. The prevalence of decreased GFR was found to be 17.3%, with a mean GFR of 75.6 mL/min/1.73m$^2$ (SE = 0.5). Table 1 presents the weighted distribution of GFR stages according to the KDIGO classification in the SABE Study of 2010 and the frequency of proteinuria in each GFR stage. It can be observed that as the glomerular filtration decreases, the frequency of proteinuria increases.

### Cumulative incidence of decreased GFR over the 6-year follow-up period and factors associated

In 2010, out of the 1,259 elderly individuals who had blood collected, 989 (82.7%) had a GFR $\geq$60 mL/min/1.73m$^2$. Among these 989 individuals, 629 (63.6%) had their blood collected again in 2016 (Fig 1). Among the elderly participants with GFR $\geq$60 mL/min/1.73m$^2$ in 2010 who were not evaluated in 2016, 161 died and 155 were lost to follow-up due to reasons such as change of address, inability to locate, institutionalization, and refusal to participate. In addition, 44 participants answered the questionnaire but did not have blood collected. When comparing the reassessed elderly with the losses, there were statistically significant differences in terms of sex, age, and education. Among the reassessed, 60.2% were female, 60.5% were aged between 60–69 years, and 34.2% had less than 3 years of education. Among the losses, 55% were female, 47.8% were aged between 60–69 years, and 38.6% had less than 3 years of education. Furthermore, 52.1% of the losses had $\geq$2 comorbidities (p = 0.18).

The cumulative incidence of decreased GFR over the 6-year follow-up period was estimated to be 14.9% (95%CI 12.3–17.9), which corresponds to an approximate annual incidence rate of 2.5%. In relation to demographic and socioeconomic factors, elderly individuals with incident decreased GFR had a higher proportion of females (67.9%), individuals aged between 70–79

**Table 1. Weighted distribution of glomerular filtration stages and frequency of proteinuria by glomerular filtration rate stage in the SABE Study of 2010.**

| Glomerular filtration rate* (mL/min/1.73m$^2$) | % | Proteinuria** % |
|---|---|---|
| $\geq$90 | 25.2 | 2.8 |
| 89–60 | 57.5 | 5.2 |
| 59–45 | 11.7 | 14.2 |
| 44–30 | 4.3 | 12.0 |
| 29–15 | 0.9 | 46.9 |
| <15 | 0.4 | 100.0 |

*Stages of glomerular filtration according to KDIGO classification.

**Proteinuria: urinary protein/creatinine ratio >0.20.

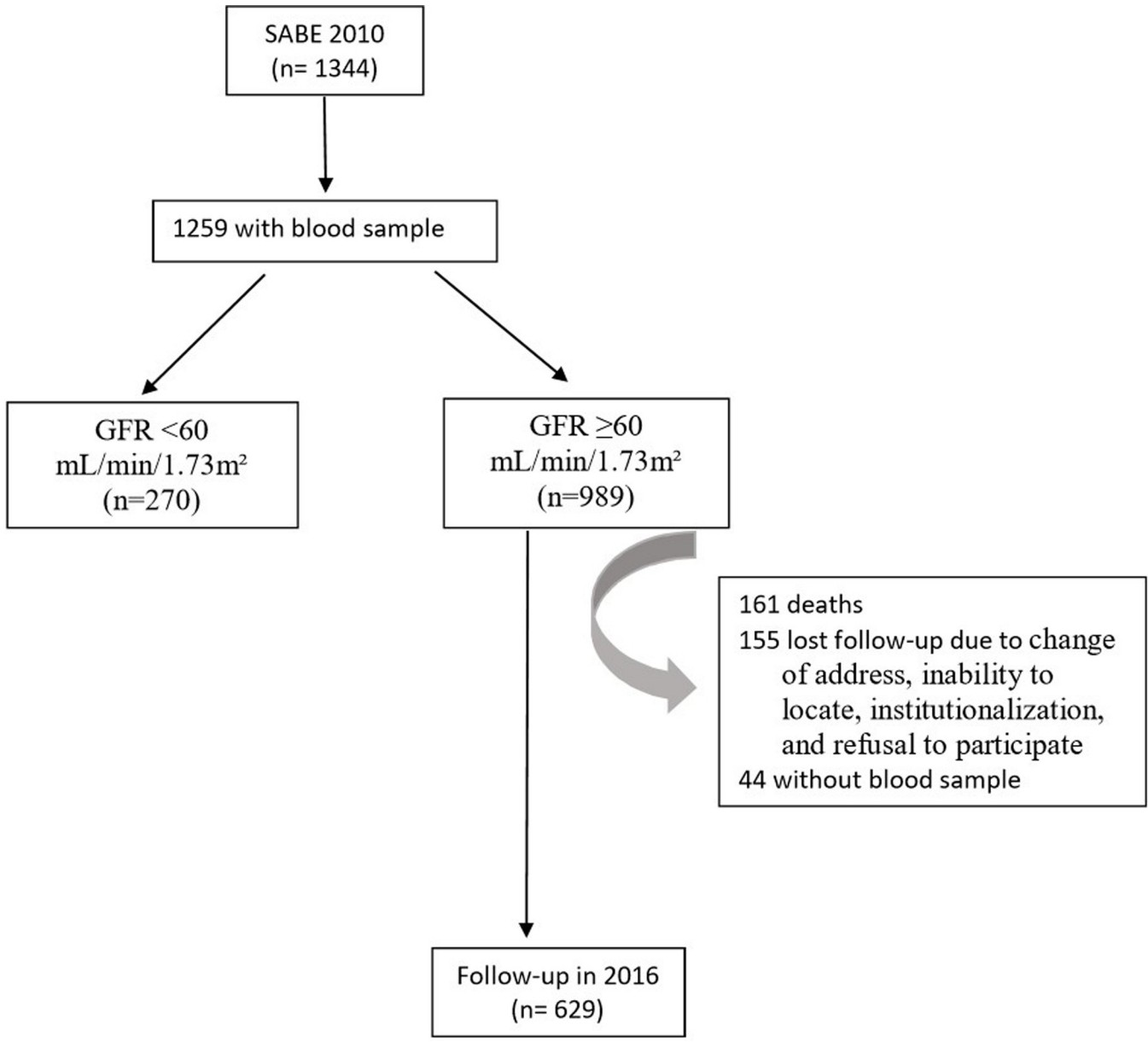

**Fig 1. Flowchart of study patients evaluated for cumulative incidence of decreased GFR estimation.**

years (44.8%), those with up to 3 years of education (44.9%), and those living with someone (88.4%). Older age and lower education were found to be associated with the incidence of decreased GFR.

Regarding health conditions and lifestyle, 62.4% of the incident cases reported having a fair/poor/very poor perception of health, 85.3% had arterial hypertension, 33% had diabetes, 67.3% had metabolic syndrome, 60.5% had 2 or more comorbidities, and 57.8% were inactive. The incidence of decreased GFR was associated with a worse perception of health, the presence of arterial hypertension, and a greater number of comorbidities. In the analysis of factors associated with the incidence of decreased GFR, comorbidities were initially analyzed separately. Elderly individuals aged 70–79 and 80 years or older had 2.3 and 3.3 times higher odds,

respectively, of having the outcome compared to those aged 60–69. Individuals with arterial hypertension had a 110% increased odds of having decreased GFR compared to those without hypertension. Elderly individuals who reported a fair/poor/very poor perception of health had a 90% increased odds of having the outcome compared to those who reported a very good/ good perception of health (Table 2).

Subsequently, the analyses were repeated, excluding the comorbidity variables and including the presence of metabolic syndrome and the number of comorbidities. In the final multiple logistic regression model, elderly individuals aged 70–79 years and those aged 80 years or older had, respectively, an increased odds of 2.3 and 3.5 times of experiencing incidents decreased glomerular filtration compared to elderly individuals aged 60 to 69 years. Among elderly individuals who had 2 or more comorbidities, there was a 2.7 times higher odds of having the outcome compared to those without any comorbidity. Furthermore, elderly individuals who reported considering their health to be fair/poor/very poor had a 80% increased odds of having the outcome when compared to those who reported considering their health to be very good/good (Table 2).

## The mean change in GFR over 6 years and factors associated

Among the 1259 elderly individuals who underwent blood collection in 2010, the mean GFR was 75.5 mL/min/1.73 m$^2$ (SE = 0.5). In 2016, a total of 743 elderly individuals repeated the

**Table 2. Crude and adjusted analysis of the incidence of decreased glomerular filtration over the 6-year study period, considering Model 1 and Model 2.** SABE Study.

| Variable | CRUDE OR (95%CI)* | ADJUSTED OR (95%CI)* |
|---|---|---|
| **MODEL 1** | | |
| **Age (years)** | | |
| 60–69 | 1.0 | 1.0 |
| 70–79 | **3.5 (2.1–5.8)** | **3.3 (2.0–5.6)** |
| 80 OR OLDER | **4.4 (2.4–8.1)** | **4.3 (2.2–8.3)** |
| **Hypertension** | | |
| NO | 1.0 | 1.0 |
| YES | **2.0 (1.1–3.6)** | **2.1 (1.1–4.0)** |
| **Perception of health** | | |
| VERY GOOD/GOOD | **1.0** | **1.0** |
| FAIR/POOR/VERY POOR | **2.0 (1.3–3.3)** | **1.9 (1.1–1.6)** |
| **MODEL 2** | | |
| **Age (years)** | | |
| 60–69 | 1.0 | 1.0 |
| 70–79 | **3.5 (2.1–5.8)** | **3.3 (2.0–5.6)** |
| 80 OR OLDER | **4.4 (2.4–8.1)** | **4.5 (2.3–8.7)** |
| **Number of comorbidities** | | |
| NONE | **1.0** | **1.0** |
| 1 | **2.6 (1.0–6.6)** | 2.6 (0.9–7.4) |
| ≥2 | **4.1 (1.7–10.3)** | **3.7 (1.3–10.5)** |
| **Perception of health** | | |
| VERY GOOD/GOOD | **1.0** | **1.0** |
| FAIR/POOR/VERY POOR | **2.0 (1.3–3.3)** | **1.8 (1.1–3.1)** |

*95%CI = 95% confidence interval.

Logistic regression model, adjusting for gender and education.

Model 1: Hosmer and Lemeshow = 0.51, Model 2: Hosmer and Lemeshow = 0.41.

blood collection, representing a sample that was representative of 796,673 elderly residents in the city of São Paulo. During this period, 269 individuals died, 189 were lost to follow-up due to changes in address, inability to locate, institutionalization, or refusal, and 58 answered the questionnaire in 2016 but did not undergo the blood test. (Fig 2) The mean GFR for these 743 elderly individuals was 77.7 mL/min/1.73 m$^2$ in 2010 and 71.3 mL/min/1.73 m$^2$ in 2016. The elderly individuals who were not reassessed in 2016 had a mean GFR of 72.0 mL/min/1.73m$^2$ in 2010. The distribution of gender, years of education, and number of comorbidities in the non-reassessed group was similar to that of the reassessed group. However, there was a statistically significant difference in terms of age, with the non-reassessed group being older compared to the group that underwent reassessment in 2016 (mean age of 72.8 versus 68.4 years, respectively). Over the 6-year period, 68.1% of the elderly experienced a worsening of GFR, while 31.9% maintained or improved their filtration. The mean change in GFR over 6 years was a decrease of 5.9 mL/min/1.73m$^2$ (SE = 0.5), indicating an approximate decline of 1 mL/min/1.73m$^2$ per year.

To assess the factors associated with worsening GFR, comorbidities were initially analyzed separately. In the final model of multiple logistic regression, elderly individuals aged 70–79 years and 80 years or older had a 100% and 60% increased odds, respectively, of experiencing GFR worsening compared to those aged 60–69 years. Additionally, individuals with diabetes had 50% increased odds of GFR worsening over the 6-year period compared to non-diabetic individuals (Table 3). Subsequently, the analyses were repeated, replacing the comorbidity variables with metabolic syndrome and the number of comorbidities. In the final multiple logistic regression model, elderly individuals aged 70–79 years had 110% increased odds of experiencing GFR worsening compared to those aged 60–69 years. Furthermore, elderly individuals with metabolic syndrome had 70% increased odds of GFR worsening compared to those without the syndrome (Table 3).

In the final linear regression model, age and diabetes mellitus (DM) were found to be associated with variation in GFR. Each 5-year increase in age was associated with a decrease of 0.72 mL/min/1.73m$^2$ in GFR during the period from 2010 to 2016. Individuals with diabetes had a decrease of 2.5 mL/min/1.73m$^2$ in GFR compared to those without diabetes (Table 4).

## Discussion

In the elderly population of the city of São Paulo, the prevalence of decreased GFR was 17.3%, and the incidence over a 6-year follow-up period was 14.9% (corresponding to an annual incidence of 2.5%), with a mean decrease of 1 mL/min/1.73m$^2$ per year. This observed incidence was higher than the incidence reported by Wu et al. (2020), who evaluated 5752 Chinese individuals aged ≥45 years and reported a 4-year incidence of decreased GFR of 2.4% calculated using the CKD-EPI creatinine equation (2009). Both ethnic and mean age differences between the Chinese study (58.8 years) and the current study (70.5 years) may partly explain this disparity. To the best of our knowledge, no other population-based study in Brazil has estimated the incidence of decreased GFR specifically in the elderly population [12].

The prevalence of decreased GFR in the elderly was 17.3%. In population-based studies conducted in Rio Branco, Acre, Brazil, and Tubarão, Santa Catarina, Brazil, both using the 2009 CKD-EPI creatinine equation, the prevalences were 13% and 13.6%, respectively [13, 14]. Unlike these studies, which were conducted in medium and small cities, the current study was conducted in a megalopolis. The positive association between GFR and urban residence has been reported in the adult population [15]. In Brazil, data are lacking concerning the prevalence of CKD, particularly among the elderly population. However, in a comprehensive survey of Brazilian adults who participated in the National Health Survey in 2013, the self-reported

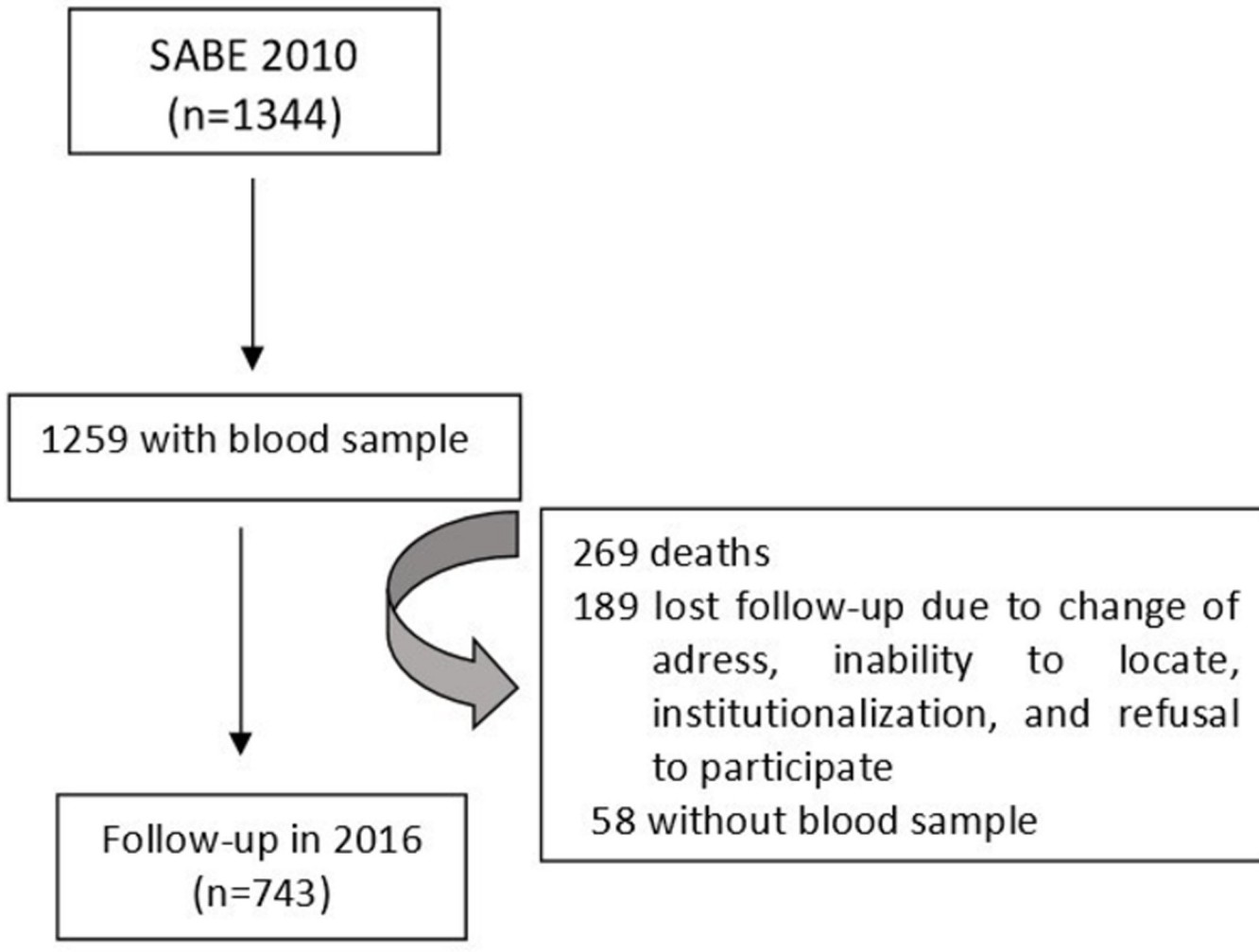

**Fig 2. Flowchart of study patients evaluated for the mean change in GFR over 6 years.**

prevalence of CKD was found to be 1.4% in the adult population and notably higher at 3.1% among individuals aged 65 years or older [16]. The estimated prevalence in the elderly population of this study was much higher than the reported rate in the general Brazilian population. It is expected that studies that use objective information such as the estimation of GFR used in this study show higher prevalence rates than those estimated solely by self-report. As CKD, or decresed GFR, has an asymptomatic onset disease, many people in the first stage do not suspect the disease and, therefore, do not seek medical diagnosis.

Several studies have suggested that the new CKD-EPI 2021 formula may potentially underestimate the GFR estimation [17, 18]. However, the consideration of race in the ethnically diverse Brazilian population presents a distinct challenge, given its susceptibility to influences of social, environmental, and cultural factors, which could contribute to biases in the study results.

The incidence and worsening of GFR over a 6-year period were found to be associated with increasing age. The kidneys naturally experience a decline in GFR with advancing age; however, socioeconomic factors and health conditions can exacerbate this process. Abdulkader et al. (2017), in a study analyzing data from the SABE Study, found that decreased GFR in the

**Table 3. Crude and adjusted analysis of factors associated with worsening glomerular filtration, considering Model 1 and Model 2.** SABE Study, 2010–2016.

| Variable | CRUDE OR (95%CI)* | ADJUSTED OR (95%CI)* |
|---|---|---|
| **MODEL 1** | | |
| **Age (years)** | | |
| 60–69 | 1.0 | 1.0 |
| 70–79 | **2.0 (1.3–3.0)** | **2.0 (1.3–3.1)** |
| 80 OR OLDER | **1.5 (1.0–2.4)** | **1.6 (1.0–2.6)** |
| **Diabetes** | | |
| NO | 1.0 | 1.0 |
| YES | **1.5 (1.0–2.2)** | **1.5 (1.0–2.3)** |
| **Quality of life(physical)** | | |
| GOOD | 1.0 | 1.0 |
| FAIR | **1.4 (1.0–2.0)** | 1.3 (0.9–1.9) |
| **MODEL 2** | | |
| **Age (years)** | | |
| 60–69 | 1.0 | 1.0 |
| 70–79 | **2.0 (1.3–2.9)** | **2.1 (1.4–3.1)** |
| 80 OR OLDER | **1.5 (1.0–2.4)** | 1.6 (0.9–2.5) |
| **Metabolic syndrome** | | |
| NO | 1.0 | 1.0 |
| YES | **1.6 (1.1–2.2)** | **1.7 (1.2–2.4)** |
| **Quality of life(physical)** | | |
| GOOD | 1.0 | 1.0 |
| FAIR | **1.4 (1.0–2.0)** | 1.4 (0.9–1.9) |

*95%CI = 95% confidence interval.

Logistic regression model adjusted for gender and education.

Model 1: Hosmer and Lemeshow = 0.26, Model 2: Hosmer and Lemeshow = 0.80.

elderly primarily occurred in the presence of other pathological conditions, suggesting that it is not solely attributable to the normal aging process [3]. Only 3.5% of elderly individuals with decreased GFR did not have other concurrent comorbidities or kidney damage, compared to 11% of those with GFR $\geq$60 ml/min/1.73m$^2$. In our study, the presence of hypertension and a greater number of comorbidities were associated with the incidence of decreased GFR, which is consistent with findings from other studies [13, 19, 20]. Hypertension is highly prevalent in the elderly population. In this study, the prevalence was 78.8%, which is consistent with the findings reported by Bouaricha, Guillé, and Puyol (2021) [21]. In their review study, they reported a prevalence of hypertension in the elderly ranging from 60 to 80%, a trend also

**Table 4. Linear regression analysis of factors associated with the variation in glomerular filtration rate.** SABE Study, 2010–2016.

| Variable | Coefficient (95%CI)* |
|---|---|
| **Age (for each 5-year increase)** | -0.72 (-1.33–0.09) |
| **Diabetes** | -2.5 (-4.83–1.00) |

*95%CI = 95% confidence interval.

Linear regression model adjusted for gender and education

observed in studies conducted in Brazil [22, 23]. Our study showed that hypertensive elderly individuals were more likely to develop decreased GFR over 6 years, highlighting the role of hypertension as a risk factor for decreased GFR. Efforts should be directed towards hypertension control, as it may potentially delay the decline of GFR. Also, renal functions among hypertensive elderly should be closely monitored. In cross-sectional studies, determining whether hypertension precedes or follows CKD is challenging, as it can act as both a cause and a consequence of decreased GFR and CKD [21]. Both conditions share common risk factors, and their association is multifactorial, involving various mechanisms of sodium dysregulation, increased sympathetic nervous system activity, and alterations in the renin-angiotensin-aldosterone system [21, 24].

Furthermore, in our study, the perception of fair/poor/very poor health was associated with the incidence of decreased GFR over 6 years, consistent with findings reported by Amaral et al. (2018) [13]. In the National Health Survey conducted in Brazil with adults aged ≥18 years, individuals who self-reported CKD were 4.7 times more likely to perceive their health as poor compared to those without CKD [14]. Surprisingly, we did not find an association between the presence of diabetes mellitus (DM) and the incidence of decreased GFR, contrary to what has been reported in other studies [1, 15, 25]. The potential for a confounding effect on these results, as well as the possibility that the sample size of incident individuals may have lacked the necessary statistical power to detect the association cannot be definitively ruled out. However, the presence of diabetes did increase by 50% the likelihood of GFR decline over 6 years. Numerous studies have established an association between diabetes and worsening GFR and CKD in the elderly [13, 15]. Diabetes induces metabolic stress through excessive glucose influx, leading to functional and structural changes in the kidneys, ultimately resulting in diabetic kidney disease (DRD) [19]. DRD is considered one of the most serious clinical outcomes of diabetes, affecting 20 to 40% of individuals with the disease [26, 27].

Metabolic syndrome (MS) was found to be associated with the worsening of GFR over a 6-year period. This finding is supported by a meta-analysis of 11 cohort studies, which showed an association between MS and a decline in renal function [28]. Another recent meta-analysis conducted with Chinese individuals aged ≥45 years revealed that those with MS had a 34% increased risk of developing CKD [12].

Among the elderly participants in our study, only 14% did not have any comorbidity, while 52% had two or more comorbidities. Individuals with two or more comorbidities had 2.7 times higher odds of developing decreased GFR over the 6-year follow-up compared to those without any associated comorbidities. This finding is consistent with previous studies that have shown an increased risk of decreased GFR and CKD with a higher number of comorbidities in the elderly population [29, 30].

The SABE study employed interviews, physical examinations, and biochemical assessments, enabling a more accurate evaluation of renal function and identification of comorbidities compared to studies relying solely on self-reported information. The SABE study was specifically designed to investigate the elderly population, in contrast to many population studies where assessments related to the elderly population are secondary analyses without a sample specifically tailored for this age group, as is the case in our study.

In our study, laboratory tests were performed only once in each cohort wave of 2010 and 2016 due to the high cost and logistical challenges involved. This is consistent with the approach taken in most population-based studies, where measurements of GFR and proteinuria are also conducted only once [13, 14]. Therefore, since we were unable to determine the presence of CKD in this study, which requires the presence of abnormalities persisting for at least 3 months, we focused on reporting the prevalence of decreased GFR in the elderly, which is a strong indication of concomitant CKD. The laboratory procedures employed in this

investigation are the most widely utilized in our nation for SCr and proteinuria measurements. For economic reasons, we were unable to access calibrated SCr in conformity with the GFR equations, although the Jeffe methods with IDMS-traceable assays mitigates biases in the GFR estimation. We did not evaluate the individual´s medication usage history. The utilization of certain medications may have a potential association with reduced GFR or the presence of proteinuria, such as nephrotoxic drugs, antidiabetic agents, or antihypertensive medications.

As expected in longitudinal studies involving elderly populations, there were substantial losses between waves, primarily due to deaths. When comparing the elderly individuals who were and were not reassessed in 2016, we observed that the losses in follow-up occurred more frequently among individuals aged 60 to 69 years, with lower levels of education, and with a higher number of comorbidities. Thus, it is possible that the estimated incidence of decreased GFR may have been underestimated, as low education and the presence of comorbidities were factors associated with its occurrence. Although these losses may introduce survival bias in the estimates, they are inevitable in longitudinal studies involving the elderly.

Our longitudinal, population-based study conducted in Brazil revealed a high frequency of GFR decline over time, resulting in an annual incidence of decreased GFR of 2.5% and a mean decrease in GFR of 1 mL/min/1.73m$^2$ per year among the elderly population.

The decline in renal function frequently manifests without noticeable symptoms, and the high prevalence of comorbidities among the studied age group exacerbates the deterioration of GFR. Continuous monitoring of renal function in the elderly is crucial for maintaining their overall health. This monitoring should be integrated into primary care settings for elderly individuals with comorbidities that can impact GFR. Understanding the factors associated with GFR progression facilitates early detection and treatment of chronic kidney disease, thereby delaying or preventing the onset of complications and ultimately leading to improved quality of life and reduced mortality rates.

## Supporting information

**S1 File. Set of data analysed for each individual included in this study.**
(XLS)

## Acknowledgments

Dirce MT Zanetta, as corresponding author, accepts responsibility for the integrity and validity of the data collected and analyzed.

## Author Contributions

**Conceptualization:** Camila de Souza dos Santos, Dirce Maria Trevisan Zanetta.

**Formal analysis:** Camila de Souza dos Santos, Dirce Maria Trevisan Zanetta.

**Funding acquisition:** Yeda Aparecida de Oliveira Duarte.

**Methodology:** Camila de Souza dos Santos, Dirce Maria Trevisan Zanetta.

**Project administration:** Yeda Aparecida de Oliveira Duarte.

**Writing – original draft:** Camila de Souza dos Santos, Dirce Maria Trevisan Zanetta.

**Writing – review & editing:** Camila de Souza dos Santos, Yeda Aparecida de Oliveira Duarte, Dirce Maria Trevisan Zanetta.

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
