## [Decision Letter · Decision Letter 0]

4 Jul 2023

PONE-D-23-15715Prevalence and incidence of decreased glomerular filtration rate and its variation over 6 years: Cohort study SABE 2010-2016PLOS ONE

Dear Dr. Santos,

Thank you for submitting your manuscript to PLOS ONE. After careful consideration, we feel that it has merit but does not fully meet PLOS ONE’s publication criteria as it currently stands. Therefore, we invite you to submit a revised version of the manuscript that addresses the points raised during the review process.

We look forward to receiving your revised manuscript.

Kind regards,

Tauqeer Hussain Mallhi, Ph.D

Academic Editor

PLOS ONE

Additional Editor Comments:

Dear Authors,

Thank you for submitting your manuscript to Plos One. Your manuscript has been evaluated by three relevant experts. They found that the manuscript carries pivotal merits but has areas of improvement in discussion. I will suggest considering the comments of the reviewers carefully to revise the draft.

Reviewers' comments:

Reviewer's Responses to Questions

**Comments to the Author**

1. Is the manuscript technically sound, and do the data support the conclusions?

Reviewer #1: Yes

Reviewer #2: Yes

Reviewer #3: Yes

2. Has the statistical analysis been performed appropriately and rigorously? 

Reviewer #1: Yes

Reviewer #2: Yes

Reviewer #3: Yes

3. Have the authors made all data underlying the findings in their manuscript fully available?

Reviewer #1: Yes

Reviewer #2: Yes

Reviewer #3: No

4. Is the manuscript presented in an intelligible fashion and written in standard English?

Reviewer #1: Yes

Reviewer #2: Yes

Reviewer #3: Yes

5. Review Comments to the Author

Reviewer #1: This is an ambitious study with the objective of the assessment of the prevalence and the incidence of the decrease glomerular filtration rate (< 60 mL/min/1.73m2) in elderly patients over a period of 6 years (2010-2015). The prevalence of decrease eGFR in 2010 was 17.3%; The incidence of decreased eGFR between 2010-2016 was 14.9% The unique modifiable factor associated with decreased eGFR was hypertension. The authors should suggest that an intervention on this factor may delay the decrease in eGFR

Reviewer #2: Thanks to all authors for such an interesting work.

I have the following points:

1. In the Introduction, Line 38- Line 42: It is not only in elderly population but even on larger population scale. It also needs a reference.

2. In the Methods, Line 71: tem - ten

3. It would be better to elaborate more on the use of Sulfosalicylic acid for proteins and Jaffe method for Cr. Please put justification in the discussion. Do you consider it a limitation?

4. Elaborate more on the use of subjective assessment and self reporting. How this could affect the results of your study?

5. How the difference between 2021 CKD-EPI GFR values affected the results for the 2010 and 2016 cohorts? Please explain in the discussion.

6. In the statistical analysis, please define and clarify what you mean by (Weighted) numbers.

7. In the results: It would be better to add a figure that describes the flow of the different cohorts and the study participants.

8. In the Discussion, you compared the incidence with Wu et al. 2020 which was a Chinese cohort and you attributed the difference to the difference in the average age. Was there any effect of the ethnicity (Chinese vs. Brazil) considering that Wu et al used the CKD-EPI 2009?

9. As the main cause for CKD in your cohort is hypertension, It would be better to mention the prevalence of Hypertension in Brazil.

10. Line 289, better to add a reference.

11. Is there any data about CKD prevalence in < 60 years old Brazilians? Please mention

12. Could you please, compare the annual incidence of decreased GFR (2.5%) with other nearby countries if available.

Reviewer #3: Dear Authors,

I have reviewed your manuscript titled, "Prevalence and incidence of decreased glomerular filtration rate and its variation over 6 years: Cohort study SABE 2010-2016" and I have the following few comments:

Topic: No comments

Introduction: No comments

Methods:

The study design and sampling are well described. With respect to identified variables, the participants medication history (especially the use of anti diabetic agents, antihypertensives and proteinuria modifying agents such as angiotensin converting enzyme inhibitors/Angiotensin receptor blockers) is not captured. This is especially relevant in this study assessing GFR decline and in which proteinuria was a independent variable in the analysis.

The statitistical methods are appropriate and robust.

Results:as quite a number of results have been presented, I would suggest having subheadings to indicate what results are about to be presented as it is currently dififcult following and maintaining the authors' train of thought in the current format.

Discussion:

As mentioned in your discussion, it is quite interesting that diabetes mellitus had no association with the incidence of renal function decline. Could this be the result of potential confounding especially as the presence of diabetes incresaed the likelihood of GFR decline by 50%?

Thank you.

6. PLOS authors have the option to publish the peer review history of their article (what does this mean?). If published, this will include your full peer review and any attached files.

Reviewer #1: **Yes: **Teresa Adragão

Reviewer #2: No

Reviewer #3: No

<quillbot-extension-portal></quillbot-extension-portal>

---

## [Author Response · Author response to Decision Letter 0]

19 Oct 2023

Dear Editor,

Thank you very much for the careful revision of our work and for the opportunity to submit a revised version of the manuscript “Prevalence and incidence of decreased glomerular filtration rate and its variation over 6 years: Cohort study SABE 2010-2016”, by Santos CS, Duarte YAO, Zanetta DMT.

All the questions and issues raised by the editor and reviewers were individually addressed bellow. 

Reviewer #1: This is an ambitious study with the objective of the assessment of the prevalence and the incidence of the decrease glomerular filtration rate (< 60 mL/min/1.73m2) in elderly patients over a period of 6 years (2010-2015). The prevalence of decrease eGFR in 2010 was 17.3%; The incidence of decreased eGFR between 2010-2016 was 14.9% The unique modifiable factor associated with decreased eGFR was hypertension. The authors should suggest that an intervention on this factor may delay the decrease in eGFR

Answer: We thank you very much for the kind words. We added the following frase (line 314) in the text, accepting your suggestions:

“Efforts should be directed towards hypertension control, as it may potentially delay the decline in GFR. Also, renal functions among hypertensive elderly should be closely monitored.”

Reviewer #2: Thanks to all authors for such an interesting work.

I have the following points:

1. In the Introduction, Line 38- Line 42: It is not only in elderly population but even on larger population scale. It also needs a reference.

Answer: We thank you very much for the kind words. We’ve removed the word “elderly” from the sentence and added a reference, accepting your suggestions.

2. In the Methods, Line 71: tem – ten

Answer: We’ve replaced the word. Thank you for noticing this typo. 

3. It would be better to elaborate more on the use of Sulfosalicylic acid for proteins and Jaffe method for Cr. Please put justification in the discussion. Do you consider it a limitation?

Answer: We included the following phrase (line 362) in the discussion:

“The laboratory procedures employed in this investigation are the most widely utilized in our nation for SCr and proteinuria measurements. For economic reasons, we were unable to access calibrated Serum Creatinine (SCr) in conformity with the GFR equations, although the Jeffe methods with IDMS-traceable assays mitigates biases in the GFR estimation.”

4. Elaborate more on the use of subjective assessment and self reporting. How this could affect the results of your study?

Answer: We accepted your suggestion and added a phrase (line 288), as follow:.

 “It is expected that studies that use objective information such as the estimation of GFR used in this study show higher prevalence rates than those estimated solely by self-report. As CKD, or decresed GFR, has an asymptomatic onset disease, many people in the first stage do not suspect the disease and, therefore, do not seek medical diagnosis.” 

5. How the difference between 2021 CKD-EPI GFR values affected the results for the 2010 and 2016 cohorts? Please explain in the discussion.

Answer: We added a the following paragraph (line 293), accepting your suggestions: 

 “Several studies have suggested that the new CKD-EPI 2021 formula may potentially underestimate the GFR estimation (INKER et al. 2021; HSU et al. 2021). However, the consideration of race in the ethnically diverse Brazilian population presents a distinct challenge, given its susceptibility to influences of social, environmental, and cultural factors, which could contribute to biases in the study results.” 

6. In the statistical analysis, please define and clarify what you mean by (Weighted) numbers.

Answer: We added the following phrase (line 102) at the beggining of the Statistical Analysis section:

“The analysis incorporated the weights of individuals to adjust for varying selection probabilities among participants, ensuring the results' representativeness for the population of São Paulo city aged 60 years or older.”

Please notice that the final paragrapg within this section explains the need to presente the results in weighted numbers. We included the last frase in this paragraph to make it clearly

“The study design and the sample complexity were taken into account, employing appropriate tests and statistical analyses suitable for survey-type studies ("svy"). Sample weights were applied to ensure the representativeness of the elderly population in the city of São Paulo. The results are presented as weighted numbers.”

7. In the results: It would be better to add a figure that describes the flow of the different cohorts and the study participants.

Answer: We thank you for this suggestion, that improved the conprehension of the patientes evaluated in the study. We included the Figures 1 and 2.

8. In the Discussion, you compared the incidence with Wu et al. 2020 which was a Chinese cohort and you attributed the difference to the difference in the average age. Was there any effect of the ethnicity (Chinese vs. Brazil) considering that Wu et al used the CKD-EPI 2009?

Answer: We agree that ethinic diferences may also contribute to the diferences observed between both studies, so we included this observation in the frase, as follow:

“Both ethnic and mean age differences between the Chinese study (58.8 years) and the current study (70.5 years) may partly explain this disparity.”

9. As the main cause for CKD in your cohort is hypertension, It would be better to mention the prevalence of Hypertension in Brazil.

Answer: We added the following phrase (line 309) in the Discussion, accepting your suggestions:

“In this study, the prevalence was 78.8%, which is consistent with the findings reported by Bouaricha, Guillé, and Puyol (2021). In their review study, they reported a prevalence of hypertension in the elderly ranging from 60 to 80%, a trend also observed in studies conducted in Brazil [22,23].”

10. Line 289, better to add a reference.

Answer: We added a reference, accepting your suggestions, now is line 318

11. Is there any data about CKD prevalence in < 60 years old Brazilians? Please mention

Answer: Data on CKD prevalence as scarse in Brasil. We added the following frase (line 282) on the Discussion section:

“In Brazil, data are lacking concerning the prevalence of CKD, particularly among the elderly population. However, in a comprehensive survey of Brazilian adults who participated in the National Health Survey in 2013, the self-reported prevalence of CKD was found to be 1.4% in the adult population and notably higher at 3.1% among individuals aged 65 years or older.” 

12. Could you please, compare the annual incidence of decreased GFR (2.5%) with other nearby countries if available. 

Answer: We were not able to find incidence of decreased GFR in other nearby countries. 

Reviewer #3: Dear Authors,

I have reviewed your manuscript titled, "Prevalence and incidence of decreased glomerular filtration rate and its variation over 6 years: Cohort study SABE 2010-2016" and I have the following few comments:

Topic: No comments

Introduction: No comments

Methods:

The study design and sampling are well described. With respect to identified variables, the participants medication history (especially the use of anti diabetic agents, antihypertensives and proteinuria modifying agents such as angiotensin converting enzyme inhibitors/Angiotensin receptor blockers) is not captured. This is especially relevant in this study assessing GFR decline and in which proteinuria was a independent variable in the analysis.

The statitistical methods are appropriate and robust.

Answer: The medication usage history of the individuals was not evaluated and we included the following sentence (line 366):

“We did not evaluate the individual´s medication usage history. The utilization of certain medications may have a potential association with reduced GFR or the presence of proteinuria, such as nephrotoxic drugs, antidiabetic agents, antihypertensive medications.” 

Results:as quite a number of results have been presented, I would suggest having subheadings to indicate what results are about to be presented as it is currently dififcult following and maintaining the authors' train of thought in the current format.

Answer: We have included the suggested subheadings. Thank you very much for this suggestion, that makes it easier to understand the results. 

Discussion:

As mentioned in your discussion, it is quite interesting that diabetes mellitus had no association with the incidence of renal function decline. Could this be the result of potential confounding especially as the presence of diabetes incresaed the likelihood of GFR decline by 50%?

Answer:We added the following sentence (line 328):

“The potential for a confounding effect on these results, as well as the possibility that the sample size of incident individuals may have lacked the necessary statistical power to detect significant differences cannot be definitively ruled out.”

---

## [Decision Letter · Decision Letter 1]

7 Nov 2023

Prevalence and incidence of decreased glomerular filtration rate and its variation over 6 years: Cohort study SABE 2010-2016

PONE-D-23-15715R1

Dear Dr. Santos,

We’re pleased to inform you that your manuscript has been judged scientifically suitable for publication and will be formally accepted for publication once it meets all outstanding technical requirements.

Kind regards,

Tauqeer Hussain Mallhi, Ph.D

Academic Editor

PLOS ONE

Additional Editor Comments (optional):

Dear Authors, thank you for revising for the manuscript.

Reviewers' comments:

Reviewer's Responses to Questions

**Comments to the Author**

1. If the authors have adequately addressed your comments raised in a previous round of review and you feel that this manuscript is now acceptable for publication, you may indicate that here to bypass the “Comments to the Author” section, enter your conflict of interest statement in the “Confidential to Editor” section, and submit your "Accept" recommendation.

Reviewer #1: (No Response)

Reviewer #2: All comments have been addressed

2. Is the manuscript technically sound, and do the data support the conclusions?

Reviewer #1: Yes

Reviewer #2: Yes

3. Has the statistical analysis been performed appropriately and rigorously? 

Reviewer #1: Yes

Reviewer #2: Yes

4. Have the authors made all data underlying the findings in their manuscript fully available?

Reviewer #1: Yes

Reviewer #2: Yes

5. Is the manuscript presented in an intelligible fashion and written in standard English?

Reviewer #1: Yes

Reviewer #2: Yes

6. Review Comments to the Author

Reviewer #1: The objective of this study was to assess prevalence and incidence of decreased glomerular filtration rate (GFR) (GFR <60mL/min/1.73m2) over a six-year period in elderly residents of São Paulo. This study relied on data from 2010 and 2016 waves of the cohort SABE Study - Health, Wellbeing, and Aging, with a probabilistic and representative sample of elderly individuals residing in São Paulo. GFR was calculated using the 2021 Chronic Kidney Disease Epidemiology Collaboration creatinine (CKD-EPI) equation.

Over the study period, 68.1% of the elderly participants experienced deterioration in GFR, with an average decline of 1 mL/min/1.73m2 each year. Renal function decline often occurs without noticeable symptoms, and the high prevalence of comorbidities contributed to the worsening of GFR.

Therefore, monitoring renal function in the elderly is crucial for effectively managing the health of this population.

This message is not new

It is, however, the result of an ambitious project

The relevance of this study relies on the large sample of patients evaluated

This article fulfills all the criteria for publication on PLos one

Reviewer #2: (No Response)

7. PLOS authors have the option to publish the peer review history of their article (what does this mean?). If published, this will include your full peer review and any attached files.

Reviewer #1: **Yes: **Teresa Adragao

Reviewer #2: **Yes: **Ala Ali

---

## [Editor Report · Acceptance letter]

20 Dec 2023

PONE-D-23-15715R1 

PLOS ONE

Dear Dr. de Souza dos Santos, 

I'm pleased to inform you that your manuscript has been deemed suitable for publication in PLOS ONE. Congratulations! Your manuscript is now being handed over to our production team.

Kind regards, 

on behalf of

Dr. Tauqeer Hussain Mallhi 

Academic Editor

PLOS ONE